# Plasticity of adult coralline algae to prolonged increased temperature and $p$CO$_2$ exposure but reduced survival in their first generation

**Tessa M. Page**[ID]**, Guillermo Diaz-Pulido**[ID]*

Griffth University School of Environment and Science and Australia Rivers Institute, Griffith University, Brisbane, Queensland, Australia

* g.diaz-pulido@griffith.edu.au

**Data Availability Statement:** All relevant data are within the paper and its Supporting Information files.

## Abstract

Crustose coralline algae (CCA) are vital to coral reefs worldwide, providing structural integrity and inducing the settlement of important invertebrate larvae. CCA are known to be impacted by changes in their environment, both during early development and adulthood. However, long-term studies on either life history stage are lacking in the literature, therefore not allowing time to explore the acclimatory or potential adaptive responses of CCA to future global change scenarios. Here, we exposed a widely distributed, slow growing, species of CCA, *Sporolithon* cf. *durum*, to elevated temperature and $p$CO$_2$ for five months and their first set of offspring (F$_1$) for eleven weeks. Survival, reproductive output, and metabolic rate were measured in adult *S.* cf. *durum*, and survival and growth were measured in the F$_1$ generation. Adult *S.* cf. *durum* experienced 0% mortality across treatments and reduced their O$_2$ production after five months exposure to global stressors, indicating a possible expression of plasticity. In contrast, the combined stressors of elevated temperature and $p$CO$_2$ resulted in 50% higher mortality and 61% lower growth on germlings. On the other hand, under the independent elevated $p$CO$_2$ treatment, germling growth was higher than all other treatments. These results show the robustness and plasticity of *S.* cf. *durum* adults, indicating the potential for them to acclimate to increased temperature and $p$CO$_2$. However, the germlings of this species are highly sensitive to global stressors and this could negatively impact this species in future oceans, and ultimately the structure and stability of coral reefs.

## Introduction

Coral reefs depend on a multiplicity of organisms to ensure their success. Crustose coralline algae (CCA) are integral inhabitants of coral reef ecosystems, building and stabilising the carbonate framework of coral reefs by deposition of calcium carbonate (CaCO$_3$) [1, 2], providing food to herbivorous fish and invertebrate grazers [1, 3], and inducing the settlement of coral larvae and other invertebrate larvae [4–6]. CCA's part in facilitating the induction of coral larvae to a reef give them a crucial role in helping coral reefs withstand and recover from disturbances, and therefore their overall resilience to future changes in our oceans [6, 7].

**Funding:** This project was funded by the Australian Research Council (DP160103071) awarded to G.D-P. https://www.arc.gov.au/ The funders had no role in study design, data collection and analysis, decision to publish, or preparation of the manuscript.

**Competing interests:** The authors have declared that no competing interests exist.

Coral reefs, which have already experienced warming and are predicted to experience more [8], are among the most threatened ecosystems by environmental changes brought on by the increase in atmospheric $CO_2$, namely the increase in ocean temperature, also termed ocean warming (OW), and ocean acidification (OA), both of which threaten key, calcifying species that make up the framework of reefs, such as CCA. The responses of CCA to changes in their environment are variable, having been found to be mostly either vulnerable or insensitive to both increasing ocean temperature and/or OA [9–12]. CCA secrete the most soluble polymorph of $CaCO_3$, high Mg-calcite [13] and their process of calcification is suggested to be biologically induced rather than controlled [14], and therefore are thought to be particularly sensitive to changes in seawater carbonate chemistry that occur with OA [15–19]. In saying this, however, there are examples of CCA taxa that are well adapted to extreme environments, such as the freshwater CCA *Pneophyllum cetinaensis* [20], and the arctic-subarctic *Clathromorphum* genus [21, 22], which is able to calcify under dark conditions [23]. Moreover, some CCA taxa, particularly those from high energy environments, have areas of the thallus which are rich in dolomite, resulting in lower dissolution rates under high-$CO_2$ treatment [23, 24]. Increases in $p$CO$_2$ have been found to negatively affect the abundance and community structure of CCA [9, 10]. While, when examining the photophysiology of CCA, the effect of $p$CO$_2$ on metabolic rate (e.g. net photosynthesis, gross photosynthesis, and respiration) shows species of adult CCA studied to be largely insensitive [11, 12], although there is considerable variability across taxa. No change or increase in $O_2$ production in response to increased $p$CO$_2$ in algae may be attributed to the presence of carbon concentrating mechanisms or CCMs [25]. A study on the CCA species *Porolithon onkodes* found gross photosynthesis and respiration to be unaffected by increased $p$CO$_2$ but net photosynthesis was negatively affected by increased $p$CO$_2$ [12]. Another study found no effect of $p$CO$_2$ on net photosynthesis, gross photosynthesis, or respiration [11]. The combined effects of increased temperature and OA have been found to decrease calcification [26] and increase mortality and dissolution [16] in adult CCA species.

Much of the work on CCA and environmental change has been done with adult algae, with fewer studies investigating the combined and independent effects of increased temperature and $p$CO$_2$ on early life history stages (i.e. spores and germlings). Early life history stages of organisms are thought to be the most influenced by changes in their environment [27, 28]. Our knowledge on offspring or early life history stages of CCA is limited and variable, with studies finding early life history stages to be highly susceptible to combined stressors of increases in temperature and OA [29–33] or largely unaffected by $p$CO$_2$ when it is an independent stressor [34]. A more recent study even suggests CCA can gain tolerance to OA over multiple generations [35]. A study done examining the combined and independent effects of increased temperature, $p$CO$_2$, and irradiance on germination success of the CCA species *Porolithon* cf. *onkodes* found elevated $p$CO$_2$ was the main driver behind a reduction in the rate of spore germination and a higher rate of abnormalities in developing spores, with the magnitude of the effect being enhanced by elevated temperature [29]. Increased $p$CO$_2$ also resulted in reduced vertical and marginal growth of germlings [29]. However, an increase in $p$CO$_2$ was found to increase gross photosynthesis and respiration of germlings from *P.* cf. *onkodes* [30]. When the effect of elevated $p$CO$_2$ was studied on the germlings from another species of coralline algae, *Phymatolithon lenormandii*, an increase in mortality and abnormalities were found under more acidic conditions, with as little as a 0.1 pH unit change negatively impacting both survival and development [33]. Population persistence of macroalgae rely on spores settling and adhering onto substratum, to continue the algal lifecycle. In a study done by Guenther et al. [31], reduced pH, or the increase of $p$CO$_2$, resulted in a 40–52% delay in spore attachment and weakened the attachment strength in the coralline alga species, *Corallina vancouveriensis*. The previously mentioned studies examined shorter-term exposure (1 month or less) to OW and OA, either in combination or

independently, and found mostly negative responses over a number of genera. However, a recent study looking at the independent effect of OA on a species of CCA, *Hydrolithon reinboldii*, found that this species can gain tolerance to OA alone over multiple generations [35], which suggests it's possible for other species of CCA and corallines alike to exhibit resistance or transgenerational acclimation to OA. It is unknown, however, if and how increased temperature will negate this, and whether other species show similar outcomes.

Here, our study sought to examine the potential for a widely distributed, slow growing, reef building species of CCA, *Sporolithon* cf. *durum* (*Sporolithon* hereafter), to acclimate to global change related stressors across lifecycles. Although *Sporolithon* is not one of the major reef building species of CCA in tropical reefs, like, but not limited to, *P. onkodes* [1], *Sporolithon* is an important reef benthic component [36] (pers. obsv) and a dominant coralline alga of rhodolith beds in the topics and subtropics, particularly in mid-to deep water environments [37–39]. *Sporolithon* spp. mineralise high Mg-calcite within their cell walls like the primary reef building CCA species [14]. Therefore, furthering our knowledge on the responses of *Sporolithon* to elevated temperature and OA [40] is not only relevant to, but vital for understanding the effects of such stressors on benthic reef communities more broadly and those species that function to cement the framework of coral reefs. Ability to acclimate to environmental conditions through phenotypic plasticity has been seen in previous studies across a range of organisms, partially or fully ameliorating the negative effects of increased temperature and $pCO_2$ [35, 41, 42]. In this study, we present data on the performance of adults (survival and metabolic rates, and reproductive output) and their $F_1$ germlings (growth and survival) from the species *Sporolithon* exposed to increased temperature and $pCO_2$ over five months and eleven weeks, respectively. Long-term ocean warming (OW) and OA experiments (months to years) are less common than short-term exposure (days to weeks) experiments [43, 44], therefore, the longevity of this experiment enables us to examine the acclimatory potential of this species. It was hypothesised that $pCO_2$ would negatively affect the growth and survival of the germlings and this negative effect would be enhanced by increased temperature. The order that *Sporolithon* belongs to, Sporolithales, has basal evolutionary origins having diverged and persisted during times of elevated temperature and $pCO_2$ (relative to current levels) [45–47], due to this, it was hypothesised that the adult *S.* cf. *durum*, although a more recently diverged species from the genus *Sporolithon*, would acclimate more readily than their germlings to prolonged exposure to increased temperature and $pCO_2$, resulting in low mortality and no effect on metabolic rate across all treatments.

## Materials and methods

### Ethics statement

A permit was obtained for the described study from the Great Barrier Reef Marine Park Authority (Permit number G18/41291.1). All collections (i.e. species collected and collection sites) were in accordance with this permit. No protected species were sampled during this study and care was taken when collecting algae to minimise any impact on the reef.

### Algae collection

Adult, reproductive fragments of *Sporolithon* were collected from reefs (14˚ 41' 16.4112" S, 145˚ 27' 55.7784" E) surrounding Lizard Island, Great Barrier Reef (GBR), Australia, in January 2018. Fragments of *Sporolithon* (~4 cm$^2$) were collected on SCUBA using hammer and chisel at ~7–9 m depth. The fragments of *Sporolithon* collected were totally crustose and occurred in a stable, low light environment (~30–50 μmol photons m$^{-2}$ s$^{-1}$). Fragments were brought back to Lizard Island Research Station (LIRS) and kept in common garden in a flow through, 300 L tank in an outdoor aquarium space. Flow through water was supplied from

LIRS aquarium system where water is taken from intake plumbing on the reef, immediately in front of LIRS, and filtered upon entrance into the common garden tanks. Each fragment was checked for reproductive structures, or sori [48], which are indicative of the tetrasporophytic reproductive stage for *Sporolithon*. The surface area of each fragment that had sori on it was kept as consistent as possible (verified and noted for each fragment under a microscope), allowing, to the best of our abilities, for similar number of reproductive structures per fragment (~30–40% of surface area covered by sori). Non-reproductive fragments, and those fragments that did not have 30–40% of their surface area covered by sori, were returned to the reef they were collected from. Reproductive fragments (n = 50) were thoroughly, but gently, cleaned of epiphytes under a microscope using a soft brush and razor blade at LIRS and then transported to the aquatics laboratory at Griffith University. *Sporolithon* were transported in individually sealed plastic bags filled (50%) with food grade $O_2$. Fragments within each bag were individually wrapped in a lab napkin dampened in seawater. After transport, algae were placed immediately into a common garden tank at a constant 27°C, 8.0 pH, and salinity of 35, mimicking seawater conditions at collection site. Seawater used here was obtained from the Gold Coast Seaway (explained further in Materials and methods section Experimental setup). Two circulating pumps (Aqua One® 8 W) provided adequate water movement in the acclimation tank. Light, measured using underwater quantum sensor LI-192 connected to the light meter LI-250A (LI-COR), was kept around 30 μmol photons $m^{-2} s^{-1}$, again, mimicking irradiance levels at collection site (30–50 μmol photons $m^{-2} s^{-1}$), which were measured using the underwater quantum sensor LI-192 on a 20 m diving cable connected to the light meter LI-250A (LI-COR). Adult fragments of *Sporolithon* were allowed to acclimate for 14 days before being placed into experimental treatments. During this time, fragments were again thoroughly cleaned and checked for epiphytes under a microscope, and presence of reproductive structures (sori) was checked using a dissecting microscope towards the end of the 14 days of acclimation. Adult fragments that were most similar in size and crust area occupied by sori were placed into individual, independent experimental tanks after 14 days in common garden.

## Experimental setup

To address the effect of increased temperature and $p$CO_2 on the reproductive output, survival, and metabolic processes of adults, and the survival and growth of $F_1$ germlings of the CCA *Sporolithon* we exposed adults and germlings to a multifactorial laboratory experiment in a purpose-built aquatics laboratory at Griffith University's Nathan Campus. An indoor, 4-header sump (200 L) recirculating seawater system with 44 (4 L) individual treatment tanks was set up in the aquatics laboratory at Griffith University. 44 fragments of *Sporolithon* were allocated into one of 44 individual, independent treatment tanks. 3000 L of filtered (5 μm filter) seawater was brought in from the Gold Coast, Seaway, collected on an incoming high tide (Gold Coast, Australia) every two weeks and stored in a primary header/holding tank with two, 3000 L $hr^{-1}$ circulating pumps to ensure adequate water movement. Seawater in the primary header/holding tank was kept at 27°C, 35 ppt salinity, and had a pH of ~8.0. Partial water changes on the experimental system (3/4 header sump volume) were done every five days to ensure ambient nutrient levels [49], while complete water changes on the experimental system were done every 15 days. pH, salinity, temperature, and alkalinity were measured prior to water changes and after water changes at the beginning of the experiment and no significant difference was found between the values. Water moved from the primary holding tank into the four header sumps was adjusted to respective treatment condition before being pumped (Pond One® Pondmaster 3600) into the 44 independent treatment tanks through separate inflow tubes for each tank (S1 Fig). Temperature and $p$CO_2 were regulated in the header

sumps at ambient (27˚C, pH 8.0; ~400 µatm), high temperature + ambient $pCO_2$ (29˚C, pH 8.0; ~400 µatm), ambient temperature + high $pCO_2$ (27˚C, pH 7.7; ~1000 µatm), and high temperature + high $pCO_2$ (29˚C, pH 7.7; ~1000 µatm). In treatments where temperature and $pCO_2$ were manipulated, they were gradually increased over seven days to reach target values, ultimately that of 0.3 units below a pH of 8.0 and +2.0˚C ambient temperature, relating to levels predicted by the IPCC under the RCP8.5 scenario for end of century levels [50]. pH was controlled by the use of pH-controllers (AquaController, Neptune Systems, USA) that injected either pure $CO_2$ or ambient air into the header sumps until desired value was reached. Temperature was maintained and manipulated by use of titanium heaters (EcoPlus, Aqua Heat, 300 W) within each sump that were set to desired temperature (27˚C or 29˚C). Flow to each individual treatment tank was maintained at 12 L h$^{-1}$ allowing for water circulation. A small, submersible pump (Aqua One$^{®}$ 6 W) was placed in each treatment tank as well to allow for adequate water movement. LED lights (MarsAqua 300 W LED, Full Spectrum) provided a 12 h light: 12 h dark photoperiod. During light period, levels were kept at ~30 µmol photons m$^{-2}$ s$^{-1}$. Light was measured every other day using the underwater quantum sensor LI-192 connected to a light meter LI-250 A (LI-COR, USA). Adult CCA were kept in their respective treatments for two weeks and examined under a microscope every third day to ensure reproductive structures were still there and spore release hadn't occurred, which was determined through the lack of visible settled germlings on the experimental tanks and microscopic confirmation of sori still on adult fragments. After two weeks in treatment, each adult fragment was placed on a single gridded, transparent acrylic plate, one plate per treatment tank, n = 11, for a total of 44 acrylic plates (8 cm$^2$, slightly sanded to increase substrate rugosity) and continued to be examined for another two weeks. Adults were kept in treatment for five months; mortality and health were monitored throughout the experiment. After one month in treatment, adults were induced to release spores, this way obtaining individuals of the $F_1$ (method explained in Materials and methods section Induction of spore release).

## Carbonate chemistry

Seawater pH was on the total scale (pH$_T$) and was measured twice daily (08:00 and 17:00) in all independent treatment tanks and header sumps for the first two months of the experiment and once daily for the remainder of the experiment using a portable pH meter (Mettler Toledo, SevenGo Duo SG98) paired with a pH electrode with integrated temperature probe (Mettler Toledo, InLab Routine Pro) that was calibrated to the total scale (pH$_T$) using Tris-HCl buffers [51] across five temperatures ranging from 26˚C to 30˚C. Total alkalinity ($A_T$) was measured every three days for the first three weeks of the experiment and then every week for the remainder of the experiment using open-cell potentiometric titrations (Mettler Toledo, T50) following standard practices (SOP) 3b [51]. Salinity was measured daily using a conductivity meter (Mettler Toledo, SevenGo pro) and adjusted with deionised (DI) water. PH$_T$, $A_T$, temperature, and salinity were used to calculate the carbonate chemistry parameters daily using the Seacarb package version 3.2.12 [52] in the statistical computing program, R version 3.5.1 (S1 Table). High Mg-calcite saturation state was calculated for a 16.4% MgCO$_3$ following method from Diaz-Pulido et al. [16].

## Induction of spore release

After the adult fragments of *Sporolithon* were in their respective treatment for one month (February–March 2018), we induced the release of spores ($F_1$) following an adapted method described in Ordoñez-Alvarez et al. [29]. To ensure we could quantify total reproductive output of adults and have germlings in experimental seawater throughout development (i.e. from their gametogenesis in the adult thalli through to spore release, settlement, germination, and growth

over 11 weeks) it was important adult fragments stay in treatment tanks at all times. Therefore, we increased the temperature across the entire system, four header sumps plus the 44 treatment tanks, by 2°C and doubled the light intensity (60 μmol photos $m^{-2} s^{-1}$) for four hours, the $pCO_2$/ pH was kept stable at respective treatment levels. After this period, light was returned to ambient and temperatures were returned to their respective treatment levels. The following day, the acrylic plate underneath each individual adult fragment was checked for spores/germlings. About half of the adults (n = 22) released spores following temperature and irradiance shock, resulting in either five or six successful individual tanks for each treatment that contained one acrylic plate with settled germlings. The adults that did not release spores were removed from treatments and sacrificed. Adults that successfully released spores were moved into separate treatment tanks so as to not further release spores onto the acrylic plates. All acrylic plates containing spores/germlings were immediately photographed (S2 Fig) using a camera (DP27 colour camera, Olympus Corporation) attached to an Olympus stereomicroscope system (SZX16). An area on each acrylic plate containing six germlings was identified to be followed for average growth and survival measurements, resulting in 5–6 individual tanks (n = 5–6) per treatment.

## Spore, germling, and adult measurements

Number of spores released from each adult was recorded (S2 Fig). 24 hours after initial induction, adults were removed from tanks containing acrylic plates. Plates were then placed in a shallow dish containing respective treatment water and photographed using previously mentioned camera and stereomicroscope setup. This was done only once so as to not recount already released spores or spores that might have been released after initial inductions. Photographs were then analysed using imageJ software (v 1.15s National Institutes of Health, USA) and the number of spores per acrylic plate were counted and recorded.

Germling survival and growth rates were measured throughout the experiment every week, from the 18th of March till the 1st of June 2018 (11 weeks). Both measurements were done on the same day over the 11 weeks of the experiment. Survival was recorded weekly through changes in colour (i.e. pigmented to white). Germling growth rate was estimated via imaging a marked area containing six individuals per acrylic plate per treatment tank weekly (S3 Fig), starting with $T_0$ and ending at $T_{11}$. Images were then processed in imageJ by finding the area of each germling and then averaging these values to get a mean growth rate per treatment tank (n = 5–6). Care was taken initially to choose germlings that settled what we considered an adequate distance apart to avoid coalescence, or the merging of two or more germlings to form one algal crust. Some abnormalities were observed in growth, but this was not quantified.

To estimate health and fitness of adults compared to offspring, we measured their metabolic rates. After five months, adults were removed from treatments (n = 6) to measure $O_2$ production and respiration on all individuals (N = 24). To determine $O_2$ production, adults were removed from tanks about 3 hrs into the 12-hr photoperiod. Each fragment was placed into individual, acrylic incubation chambers (~150 mL). Chambers were placed in a water bath to maintain constant temperature during incubations and above a magnetic stir plate. Chambers were equipped with a temperature sensor (Pt100), a PreSens dipping $O_2$ optode (DP-PSt3) and stir bar to ensure water movement throughout incubation. Percent $O_2$ was measured every minute over 2 hrs with the dipping $O_2$ optodes connected to a 10-channel trace $O_2$ meter (OXY-10 SMA trace, G2, PreSens, Germany). $O_2$ sensors were calibrated every morning prior to measurements to 100% $O_2$ (DI water bubbled with air) and 0% $O_2$ obtained through addition of sodium dithionite ($Na_2S_2O_4$) to DI water. Irradiance was kept similar to irradiance held throughout the experiment, 30 μmol photons $m^{-2} s^{-1}$. A chamber containing no fragment of CCA was included in each incubation to serve as a blank. Respiration was measured after incubating fragments of CCA for 1 hr in total darkness.

Respiration was measured over the following hour using the same method as $O_2$ production, but, CCA were kept in dark during measurements. $O_2$ production and respiration measurements were adjusted from a blank and normalised to ash-free dry weight [53] of each fragment.

## Data analyses

All data were analysed using the statistical computing program, R (v 3.5.1). Germling response data were tested for normality and variance using QQ normality plots. Photosynthesis, or $O_2$ production, and respiration data of adults were tested for normality using Shapiro Wilk's tests and Levene's test for homoscedasticity. If data were found not to be normal, it was log transformed to meet assumptions of normality. Two-way ANOVAs with temperature (2 levels, 27 and 29˚C) and $pCO_2$ (2 levels, ~400 and 1000 µatm) set as fixed factors [54], were run on data to determine effect of treatment and their interactions on response variables. When a significant interaction was found between elevated temperature and $pCO_2$ a t-test was performed to calculate the differences between treatments. To analyse survival, the survival R package (v 2.44–1.1) was used in R, which uses the non-parametric Kaplan-Meier estimator to estimate survival function and the Fleming-Harrington G-rho family to compare survival curves. Kruskal-Wallis chi-squared tests were done to test significant differences in survival probability across treatments.

## Results

### Reproductive output

There was no significant effect of increased temperature or $pCO_2$ (low pH) on the reproductive output of adult *Sporolithon* (p > 0.05, S2 Table and Fig 1).

### Germling survival

There was a significant decrease in survival probability amongst *Sporolithon* germlings ($F_1$) after exposure to the combined treatment of elevated temperature and $pCO_2$ after 11 weeks when compared to all other treatments (Kruskal Wallace Chi-squared = 13.5, p < 0.01, Table 1). No significant effect on survival probability was found between the remaining treatments of ambient, increased temperature, or elevated $pCO_2$ (Table 1 and Fig 2).

### Germling growth

Growth of germlings, through changes in area, was measured every seven days throughout the entirety of the experiment. After one week in treatment, there was a reduction in average area under the elevated temperature and $pCO_2$ treatment, however, this was not significant ($F_{1,20}$ = 0.001, p = 0.976, Table 2 and Fig 3). Mean area was not significantly impacted by a stressor until week seven, where increased temperature significantly reduced, ~20% decrease, the growth of germlings under the high $pCO_2$ treatment ($F_{1,19}$ = 4.928, p = 0.039, Table 2 and Fig 3). A significant effect of temperature on average growth was seen at week eight ($F_{1,19}$ = 4.606, p = 0.045, Table 2 and Fig 3), again with about a 20% reduction. At week 11, temperature again was found to be the leading influencing stressor, but its effect significantly depended on the level of $pCO_2$ (i.e. a significant interaction between temperature and $pCO_2$, $F_{1,15}$ = 0.029, p = 0.007, Table 2 and Fig 3). Here, elevated temperature caused the largest decline in germling size but only in the high $pCO_2$ treatment (reduction of ~40%) (t-test; t = 2.6, p = 0.036).

### Adult metabolic rate and survival

The mean metabolic rates ($O_2$ production and respiration) were measured on adult fragments of *Sporolithon* after five months in treatment. *Sporolithon* mean rates of $O_2$ production varied

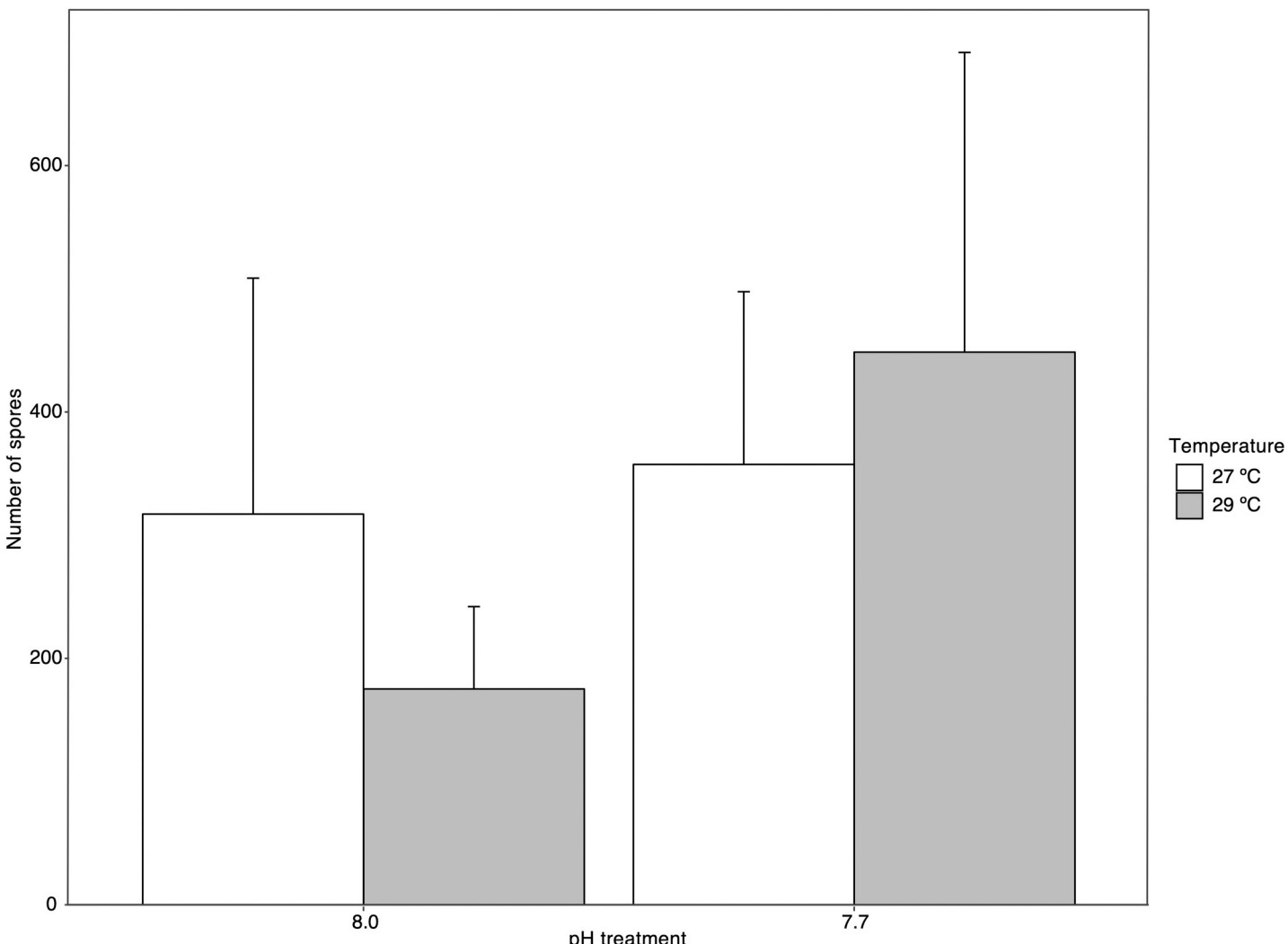

**Fig 1. Effects of temperature and $p$CO$_2$ (pH) on the reproductive output of adult *Sporolithon* after one month in treatment conditions.** Each bar represents mean number of spores released for each treatment ± SE (standard error) for n = 5–6.

between 1.74 and 2.30 μmol O$_2$ g$^{-1}$ h$^{-1}$, and respiration values ranged from 4.23 to 3.25 μmol O$_2$ g$^{-1}$ h$^{-1}$ consumed. O$_2$ production decreased with increasing $p$CO$_2$ (lowering pH, Table 3 and Fig 4), and the effect of $p$CO$_2$ on photosynthesis was found to be significant ($F_{1,18}$ = 9.56, p = 0.006, Table 3). Temperature did not have an effect on O$_2$ production ($F_{1,18}$ = 0.439,

**Table 1. Germling survival probability and R statistical analysis of survival using Kruskal-Wallis rank sum tests to determine significant differences in survival at varying treatment conditions.**

| Treatment | Observed deaths | Expected deaths | (O-E)$^2$ | (O-E)$^2$ V$^{-1}$ | Chi$^2$ d.f. = 3 | Sig. |
|---|---|---|---|---|---|---|
| pH 8.0 + 27˚C | 16 | 18.61 | 6.76 | 0.52 | n.d. | n.s. |
| pH 8.0 + 29˚C | 13 | 22.40 | 88.36 | 6.09 | n.d. | n.s. |
| pH 7.7 + 27˚C | 20 | 19.50 | 0.25 | 0.02 | n.d. | n.s. |
| pH 7.7 + 29˚C | 28 | 16.50 | 132.25 | 11.14 | 13.50 | ** |

n.d. there is no Chi-squared value because there were no differences in observed mortality across treatments. ** < 0.01.

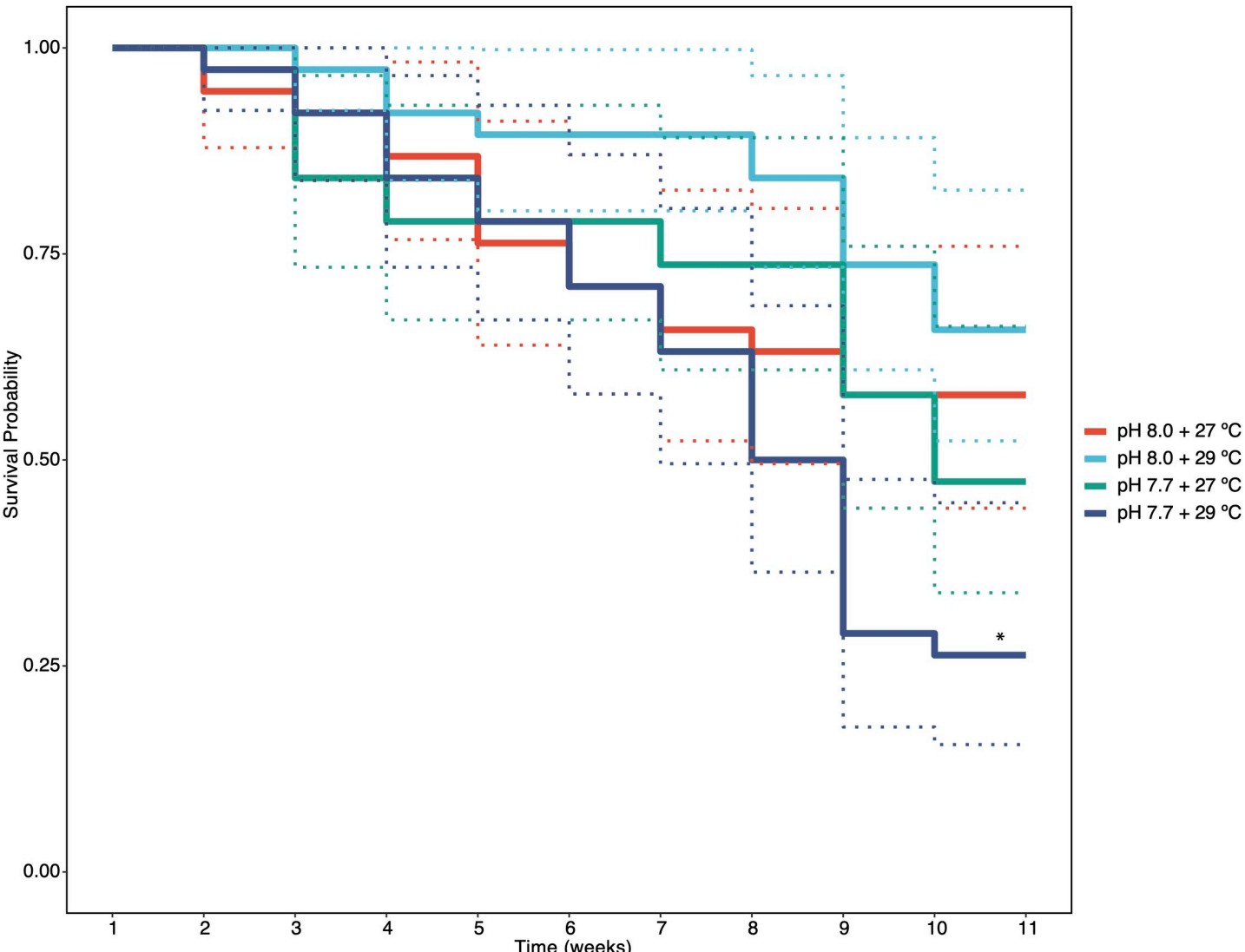

**Fig 2. Survival probability of *Sporolithon* germlings over the duration of the experiment (11 weeks) across temperature and *p*CO₂ (pH) treatments.** Solid lines represent mean values, and dotted lines represent 95% confidence intervals. The asterisk indicates a statistically significant difference at p < 0.05.

p > 0.05, Table 3), and there was no interactive effect between increased temperature and $p$CO$_2$ ($F_{1,18}$ = 0.671, p > 0.05, Table 3). There was no effect of treatment on respiration rates of *Sporolithon* (p > 0.05, Table 3 and Fig 4).

Survival in adults was not impacted by increased temperature, $p$CO$_2$, or the combination of the two after five months in treatment. There were no observed deaths throughout or at the end of the five months.

## Discussion

Our study shows a contrasting response in life history stages to environmental stressors, with adult *Sporolithon* being remarkably resistant and early life history stages being very sensitive to OW and OA. These data contribute essential knowledge to understanding the longer-term effects of global change stressors across lifecycles of an important and abundant species of

**Table 2. Results of two-way analyses of variance (ANOVA) to test the effects of increased temperature and $p$CO$_2$ (decreased pH) on the average growth of *Sporolithon* germlings.**

| Time point | Source of variation | Df | MS | F | p |
|---|---|---|---|---|---|
| Week 2 | Temperature | 1 | 0.043 | 1.735 | 0.482 |
| | $p$CO$_2$ | 1 | 0.013 | 0.516 | 0.205 |
| | Temperature * $p$CO$_2$ | 1 | 0.000 | 0.001 | 0.976 |
| | Residuals | 20 | 0.025 | | |
| Week 3 | Temperature | 1 | 0.190 | 1.860 | 0.319 |
| | $p$CO$_2$ | 1 | 0.107 | 1.047 | 0.189 |
| | Temperature * $p$CO$_2$ | 1 | 0.190 | 1.865 | 0.188 |
| | Residuals | 20 | 0.102 | | |
| Week 4 | Temperature | 1 | 0.037 | 0.515 | 0.348 |
| | $p$CO$_2$ | 1 | 0.067 | 0.924 | 0.481 |
| | Temperature * $p$CO$_2$ | 1 | 0.039 | 0.540 | 0.471 |
| | Residuals | 20 | 0.072 | | |
| Week 5 | Temperature | 1 | 0.150 | 4.076 | 0.117 |
| | $p$CO$_2$ | 1 | 0.101 | 2.732 | 0.060 |
| | Temperature * $p$CO$_2$ | 1 | 0.019 | 0.529 | 0.477 |
| | Residuals | 20 | 0.037 | | |
| Week 6 | Temperature | 1 | 0.062 | 1.650 | 0.105 |
| | $p$CO$_2$ | 1 | 0.109 | 2.911 | 0.215 |
| | Temperature * $p$CO$_2$ | 1 | 0.030 | 0.796 | 0.384 |
| | Residuals | 20 | 0.038 | | |
| Week 7 | Temperature | 1 | 0.133 | 4.928 | 0.039* |
| | $p$CO$_2$ | 1 | 0.041 | 1.528 | 0.232 |
| | Temperature * $p$CO$_2$ | 1 | 0.001 | 0.020 | 0.889 |
| | Residuals | 19 | 0.027 | | |
| Week 8 | Temperature | 1 | 0.005 | 4.606 | 0.045* |
| | $p$CO$_2$ | 1 | 0.169 | 0.147 | 0.706 |
| | Temperature * $p$CO$_2$ | 1 | 0.001 | 0.002 | 0.964 |
| | Residuals | 19 | 0.037 | | |
| Week 9 | Temperature | 1 | 0.016 | 0.612 | 0.446 |
| | $p$CO$_2$ | 1 | 0.014 | 0.544 | 0.472 |
| | Temperature * $p$CO$_2$ | 1 | 0.009 | 0.338 | 0.570 |
| | Residuals | 16 | 0.026 | | |
| Week 10 | Temperature | 1 | 0.046 | 1.811 | 0.200 |
| | $p$CO$_2$ | 1 | 0.019 | 0.758 | 0.399 |
| | Temperature * $p$CO$_2$ | 1 | 0.001 | 0.020 | 0.890 |
| | Residuals | 15 | 0.026 | | |
| Week 11 | Temperature | 1 | 0.194 | 6.758 | 0.020* |
| | $p$CO$_2$ | 1 | 0.025 | 0.881 | 0.363 |
| | Temperature * $p$CO$_2$ | 1 | 0.278 | 9.697 | 0.007** |
| | Residuals | 15 | 0.029 | | |

MS = mean square, df = degrees of freedom, $F$ = F-ratio. Note: T-test testing the effect of temperature within elevated $p$CO$_2$ at week 11 showed p = 0.0362; while the effect of temperature was not significant within ambient $p$CO$_2$ (p = 0.061). * < 0.05, ** < 0.01.

CCA and investigates the potential for acclimation in adult *Sporolithon* and survivorship and growth of germlings of the first generation (F$_1$). Adult *Sporolithon* ultimately did well throughout the entirety of this experiment, showing no mortality, no significant change in

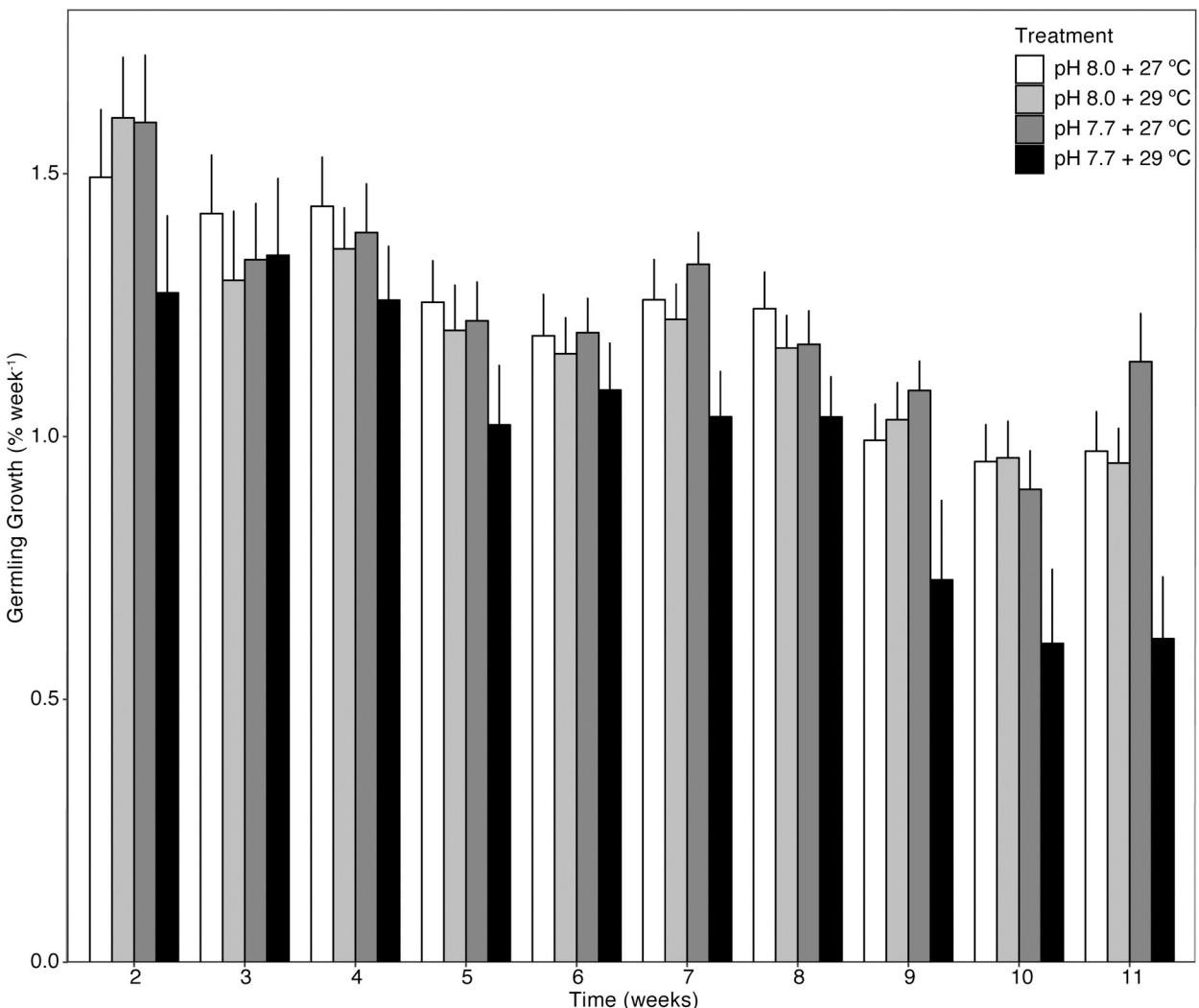

**Fig 3. The percent growth of *Sporolithon* germlings across temperature and $pCO_2$ (pH) treatments from week to week over 11 weeks.** Each bar represents mean % growth from each treatment at each time point ± SE. Data were log transformed to meet normality.

reproductive output or respiration, and photosynthesis was slightly decreased in response to increased $pCO_2$ (low pH). The results from the adult *Sporolithon* support the hypothesis that this species is robust, and the divergence time of the genus of *Sporolithon*, and more broadly

**Table 3. Results of two-way ANOVA for the effects of temperature and $pCO_2$ (pH) on the metabolic rates of adult fragments of *Sporolithon* after 5 months in treatment.**

| | Oxygen production ($\mu$mol $O_2$ g$^{-1}$ h$^{-1}$) | | | | Respiration ($\mu$mol $O_2$ g$^{-1}$ h$^{-1}$) | | |
|---|---|---|---|---|---|---|---|
| Two-way ANOVA | Df | MS | F | p | MS | F | p |
| Temperature | 1 | 0.056 | 0.439 | 0.516 | 3.026 | 1.830 | 0.192 |
| $pCO_2$ | 1 | 1.217 | 9.560 | 0.006** | 0.495 | 0.299 | 0.591 |
| Temperature * $pCO_2$ | 1 | 0.085 | 0.671 | 0.423 | 0.113 | 0.068 | 0.796 |
| Residuals | 18 | 0.127 | | | 1.654 | | |

Oxygen produced or consumed ($\mu$mol O2 g$^{-1}$ h$^{-1}$) was normalised to the ash-free dry weight of individual fragments, n = 5–6.

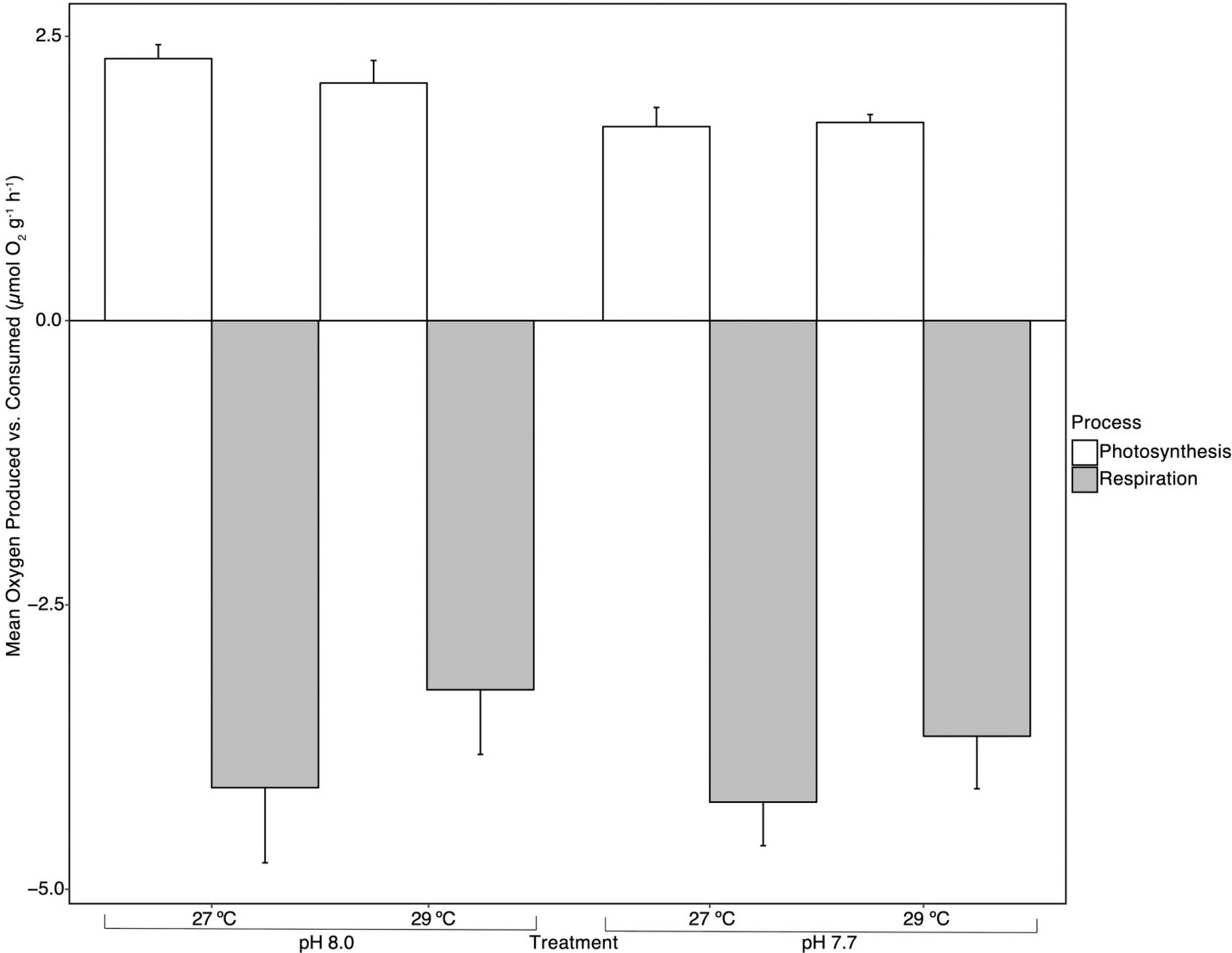

**Fig 4. Effects of temperature and $p$CO$_2$ (pH) on the mean O$_2$ production and consumption (respiration rate) of adult *Sporolithon* after five months in treatment.** Each bar represents mean O$_2$ produced or consumed ± SE, n = 5–6.

the family, may play a role in this [45–47]. The slight decrease in O$_2$ production in response to increased $p$CO$_2$ without any impact on survivorship indicates a potential acclimatory or plastic response to new CO$_2$ conditions [55, 56].

The success of adult life stages is dependent on the health of their earlier life history stages. Here, the response of *Sporolithon* germlings to global change variables may present an issue for their subsequent life history stages. There were strong and complex interactions between temperature and $p$CO$_2$. In fact, elevated $p$CO$_2$ may benefit growth, compared to low $p$CO$_2$, but, when combined with elevated temperature, there was a strong interaction leading to reduced survival and growth under the combined environmental stressors. Although the genus *Sporolithon* is not a primary reef builder, they have a similar calcification process to reef building CCA species (e.g. Nash et al. [14]) and are important members of the benthic community in tropical reefs, and the negative response of the germlings to combined environmental stressors may impact some essential functions of CCA in coral reefs under future global change scenarios.

## Adult survival & metabolic rates

After five months of exposure to elevated temperature and $p$CO$_2$, adults of *Sporolithon* survived and remained visibly healthy, with no thallus mortality observed, across all treatments. However, O$_2$ production was significantly reduced under the high $p$CO$_2$ treatments at both ambient and high temperature, with an average reduction of 25.22%. The lack of mortality and the reduction in O$_2$ production may be interpreted as a plastic or possibly an acclimatory response to prolonged exposure to elevated $p$CO$_2$, in which the alga downregulates the photosynthetic activity without significant implication for survivorship [56, 57]. This response has been commonly observed in terrestrial plants, particularly when plants experience nutrient limitation [56, 57]. Photosynthetic acclimation to elevated CO$_2$ may be attributed to an overabundance of additional carbohydrates that photosynthesis in elevated CO$_2$ provides, suppression of nitrogen assimilation, or at elevated CO$_2$ a reallocation of resources occurs within the photosynthetic apparatus, requiring less Rubisco for photosynthesis [55, 56, 58, 59]. Like in terrestrial plants, nutrients may have played a role in the acclimatory response of *Sporolithon* to prolonged increased $p$CO$_2$, and had nutrients been added, we may have seen a "positive" or increase in metabolic rate instead of a "negative" or acclimatory response (as seen in terrestrial plants) [57, 59], however, this hypothesis needs experimental testing. This is a realistic possibility as anthropogenic nutrification is occurring on reefs due to increased agricultural runoff and changing land-use [60, 61]. We acknowledge there are more ways of acclimation to increased $p$CO$_2$, however, suppression of O$_2$ production, seen in this study, in response to prolonged $p$CO$_2$ exposure in *Sporolithon* is likely their acclimatory response, similar to terrestrial plants, in which resources and activities are modulated in order to benefit the plant [56–58, 62].

If observing only the effect of $p$CO$_2$ on the O$_2$ production of species of coralline algae, contrasting responses have been seen, where elevated $p$CO$_2$ alone hasn't influenced significant changes in O$_2$ production. This was observed in the tropical CCA species, *P. onkodes*, after 14 days in treatment there was no effect of elevated $p$CO$_2$ on stable environment samples [12]. A lack of strong response was also seen in three temperate coralline algae species when exposed to increasing $p$CO$_2$ [53]. *Sporolithon* reacted conversely to the above studies, showing a decrease in net production driven by increased $p$CO$_2$, most likely as an acclimatory response to sustained increased $p$CO$_2$. The CCMs most likely present in *Sporolithon*, like in many other macroalgae [63–65], may also play a role in why a positive response, or increase in photosynthesis, was not seen in adult *Sporolithon*. Algae maintaining high affinity CCMs are predicted to not respond to or benefit considerably from increased $p$CO$_2$ in terms of metabolic processes [25]. Ultimately, the lack of mortality and decrease in photosynthesis suggests this species of CCA is quite robust and able to acclimate to changes in their environment (e.g. increased temperature and $p$CO$_2$), suggesting a positive outlook for their persistence and function in coral reef ecosystems under future global change scenarios.

## Reproductive output

Reproductive output is an essential fitness measurement across organisms and a reduction in this process could negatively impair the persistence of species. However, in this study reproductive output was not found to be significantly influenced by experimental treatment, with only a slight increase in mean number of spores released under the most stressful treatment (high temperature + high $p$CO$_2$) when compared to the ambient treatment. To our knowledge, no past studies have looked into the reproductive output of CCA after exposure to warming and acidification, however, reproductive output from other calcifying macroalgae [66] and seagrasses [67, 68] has been measured in response to OA. For a species of coralline algae, *Arthrocardia corymbosa*, altered pH did not significantly alter reproductive output [66].

Elevated $p$CO$_2$ has been found to increase reproductive output of seagrass [67, 68], and it is suspected that seagrass will be impacted by the interaction of elevated temperature and $p$CO$_2$ [67]. The slight increase of number of spores in the combined high temperature and $p$CO$_2$ treatment could have been a stress response and final push to release as many spores as possible after the additional stress of a temperature spike to induce spore release. CCA can reproduce asexually or sexually and maintain different reproductive structures based on this. For *Sporolithon* the asexual (tetrasporophytic) reproductive structures, sori, are much smaller than the general conceptacle, or asexual reproductive structure on other CCA species [69], which are sometimes even visible by the naked eye [48]. Due to their small size and the differing amount of area sorus take up on each individual crust [69], there was likely variability from adult to adult in each treatment and that may have influenced the variable response in reproductive output across treatments. Although the sori area was kept as constant as possible, large variability was inevitable. Additionally, there is no, non-invasive way to count the number of sori that contain tetraspores. We acknowledge the limitation in completely quantifying reproductive output by measuring number of spores released, and this should be further investigated. Despite the limitations and the variability observed in *Sporolithon* spore release across treatments, reproductive output was not negatively influenced by temperature and/or CO$_2$ factors, suggesting that the reproductive process is robust to warming and acidification.

### Germling survival & growth

The sensitivity of the early life history stages to the combined effects of environmental change stressors supports multiple studies that have found early life history stages to be more vulnerable to global change stressors than adults of the same species [27]. In the current study, high $p$CO$_2$ in combination with increased temperature resulted in a significant decrease in both survival probability and germling growth, consistent with past studies on the germination success and growth of germlings from the CCA *Porolithon* cf. *onkodes* [29].

Coralline algae species differ in their responses to environmental change [9–12] and this is most likely consistent across their different lifecycles as well. Similarly, other early life history stages of different taxa are highly sensitive to exposure to elevated temperature and $p$CO$_2$, with reduction in survival found in the juvenile bivalve, *Argopecten irradians*, after 45 days in treatment [70], and a total die off in the larvae of the brittle star, *Ophiothrix fragilis*, at reduced pH (7.9 and 7.7) after only 8 days in treatment [71]. Early life history stages of the coral *Porites panamensis* [72] and the tropical sea hare *Stylocheilus striatus* [73] had reduced growth under simultaneous high temperature and $p$CO$_2$. Responses of early life history stages of *Sporolithon* found in the current study suggest a similarly found [70–73] sensitivity to environmental change that could ultimately impact their persistence in future oceans.

Decreased growth rate in our *Sporolithon* germlings, particularly under combined high temperature and $p$CO$_2$, could be linked to skeletal dissolution, as documented in other CCA species [29] and a variety of marine calcifiers across life history stages [27, 74, 75]. Skeletal dissolution may be partially explained by undersaturation of seawater with respect to high Mg-calcite under high $p$CO$_2$ and high temperature (S1 Table, Ω high Mg-calcite = 0.79), although the high $p$CO$_2$ but low temperature treatment also had a Ω high Mg-calcite < 1, yet germlings experienced comparable growth rates to ambient $p$CO$_2$. Declined Ω high Mg-calcite in the calcifying fluid (as induced by low pH) has also been proposed to influence calcification in adult *Sporolithon* [40] and may be a possibility in our experimental germlings. It is also worth considering the influence of not only the carbonate chemistry (i.e. omega) of the experimental seawater and calcifying fluid on germling calcification and growth, but also the interactions of the germlings with the benthos. Germlings were grown on an acrylic substrate in the laboratory

environment, whereas, in the field germlings would most likely be growing on porous carbonate substrate with diverse endolithic and epilithic algal communities that modify carbonate chemistry via metabolic processes. Past studies have found substrate to not be critical for CCA community structure [76], however, if substrate plays a role in germling calcification and growth in the presence of lowered pH has not been investigated and it is possible that the carbonate substrate in the field could act as a buffer to changes in carbonate chemistry surrounding the germling. It is important to note that under all treatments, there was mortality and reductions in growth rate. However, the survival probability was still significantly reduced in the combined stressor treatment, whereas the other treatments maintained similar survival probability. Additionally, this can be said with growth, seeing a significant increase under increased $pCO_2$ and a significant decrease in the combined stressor treatment. The negative response of the $F_1$ germlings to the combined stressor treatment suggests a moderately low likelihood for *Sporolithon* germlings to exhibit a plastic or an acclimatory or adaptive response to combined future environmental changes, in turn greatly reducing the success of the offspring of this species in future oceans. However, the results in the present study somewhat contradict findings from a multigenerational study done with CCA species *H. reinboldii*, where juvenile resistance to OA was found, recorded as a lessening of difference in growth across generations [35]. In this study, the second-generation of CCA grew 56.1% more slowly across OA treatments relative to the ambient/control treatment, whereas the final generation (7) had only a 0.6% difference in growth when comparing OA treatments to the ambient/control treatment, indicating that sustained exposure to OA across multiple generations enables *H. reinboldii* to gain a tolerance [35]. The current study suggests that *Sporolithon* germlings may not be able to exhibit resistance to global change stressors, unlike juveniles of *H. reinboldii* [35], however, our results may have been more similar should this study have been conducted over multiple generations, and only investigating the response to OA. Alternatively, differences in germling responses between *H. reinboldii* and *Sporolithon* may be attributed to species-specific optima for calcification and growth [40].

The data in this study suggest that temperature is a critical factor influencing survival and growth of *Sporolithon* germlings. On the other hand, the increase of $pCO_2$ in isolation of the increase in temperature (i.e. in ambient temperature conditions) enhanced germling growth. As indicated earlier, however, when combined with increased temperature, growth was significantly decreased. Increased $pCO_2$ at ambient temperature has been found to positively affect growth of some algal species [77], possibly by alleviating $CO_2$ limitation of photosynthesis, or downregulation of CCMs [25]. On the other hand, studies have shown negative growth responses of some coralline algal species to elevated $pCO_2$ at ambient temperature (e.g. *P.* cf. *onkodes* recruits/germlings [30] and *Arthrocardia corymbose* [78]). In our study, temperature played a role in decreasing growth, which has been previously observed in a temperate species of coralline algae [79]. However, in a study with a tropical CCA species, *P.* cf. *onkodes*, temperature had a positive effect on growth rate [29]. The differences in responses to the independent stressors of temperature and $pCO_2$ between *P.* cf. *onkodes* and *Sporolithon* could be attributed to a number of reasons including differences between the divergence times of these genera [47, 80], their habitat preferences [81], or other unknown factors. The genus *Sporolithon* diverged during previous times of high temperature and $CO_2$, possibly suggesting that if more $CO_2$ is present, *Sporolithon* germlings will positively respond as energy usage for $CO_2$ uptake is reduced and the germlings are able to utilise the additional $CO_2$ instead of relying on CCMs. Of course, increased growth would be highly dependent on temperature (and possibly nutrients as discussed earlier), but it is likely that this increased growth may be realised in habitats exposed to comparatively lower temperatures and light regimes, such as relatively deep-water

reefs. Our study illustrates the considerable variability in germling responses to global change among CCA species (e.g. *P. onkodes* [29, 30], *H. reinboldii* [35], *P. lenormandii* [33]).

## Concluding remarks

The results from this study show polarising responses from adult *Sporolithon* and their $F_1$ germlings to increased temperature and $p$CO$_2$, suggesting life history stage plays an important role in resilience to global change variables and ability to exhibit a plastic response. Adult *Sporolithon* were largely resilient to increased temperature and $p$CO$_2$ suggesting a robust potential for acclimation, whereas the results from the $F_1$ germling growth and survival suggest that the germlings of *Sporolithon* are more sensitive to changes in their environment. The mechanisms explaining the different acclimatory response between life history stages are not well understood, but may be related to 1) variable carbon uptake physiologies (e.g. with germlings lacking or having low affinity CCM as growth increased under high $p$CO$_2$ alone); 2) variable skeletal mineralogy (e.g. nature, composition, or skeletal protection by living tissue); 3) different composition and/or quantities of cell-wall organic constituents, or 4) different nutrient requirements. In particular, CCA tissue organisation varies throughout ontogenesis [82, 83] and it is likely that different tissues that develop following germination to the formation of adult, thick crusts may have different susceptibilities to pH reduction. For example, CCA spores mainly germinate developing a basal hypothallial tissue [82] and recent studies show the hypothallus of a number of CCA species has different mineralogy from mature (perithallial) tissue (e.g. in *Phymatolithon*, perithallial cell walls have mean 13.4 mol% MgCO3 while the hypothallus has 17.1 mol% MgCO3) [13], and this may influence early germling development. There is very little information on *Sporolithon* germination [84] and further work should elucidate the contribution of these processes to the different responses to climate stressors between life history stages of CCA.

Our study also suggests a mechanism by which adult *Sporolithon* is able to cope with ocean acidification and warming conditions. A decline in $O_2$ production (with no thallus mortality) is a possible indicator of an acclimatory response in this species, and this observation has not been documented previously in other coralline algae [43, 85], but is common in some terrestrial plants [55, 56]. Although there may be a positive outlook for adult *Sporolithon* to acclimate and possibly eventually adapt to future ocean conditions, the sensitivity of *Sporolithon* germlings to combined environmental stressors complicates this species persistence in future oceans. A previous study showed OA considerably reduced net calcification of *S. durum* rhodoliths, suggesting adults of this species may also be sensitive to pH decline [40]. Although we did not measure calcification in our *Sporolithon*, the negative net calcification rate (i.e. dissolution) observed in the rhodoliths could have been due to the use of this growth form (i.e. rhodoliths, as opposed to crusts), which have considerable internal porosity, potentially facilitating skeletal dissolution. Future studies are needed to determine the effects of OW and OA on gross and net calcification of *Sporolithon* to obtain a more complete picture of the adult's sensitivity to climate stressors. Our study began as a novel attempt to conduct a multi-generational study with a slow growing, reef building species of CCA. However, multiple generations were not achieved (cf. Cornwall et al. [35]), therefore, the experiment was adapted to look at the long-term effects of global change factors across lifecycles. Acclimation history of adults prior to collection may have played a role in responses seen in this study, therefore future studies are necessary to more fully compare response to global change across life history stages. These future studies should also further investigate the adaptation potential of this slow growing, ancient group of CCA, specifically looking at the possibility for gained tolerance to both increasing temperature and OA.

## Supporting information

**S1 Fig. Diagram of recirculating, purpose-built aquarium system.** Diagram shows basic experimental setup including primary holding tank, treatment header sumps (n = 4), and independent treatment tanks (11 tanks per treatment).
(TIF)

**S2 Fig. $F_1$ germlings settled on an acrylic plate from the high temperature (+2.0°C) and ambient $p$CO$_2$ (pH) treatment.**
(TIF)

**S3 Fig. Month-old _Sporolithon_ germlings ($F_1$) under stereomicroscope from ambient temperature and high $p$CO$_2$ (low pH) treatment.** Letters identify individuals that were tracked throughout the experiment for survival and growth.
(TIF)

**S1 Table. Seawater carbonate system parameters.** Asterisk indicates parameters that were calculated using R package seacarb (v 3.2.12). High Mg-calcite was calculated for 16.4% calcite following methods from Diaz-Pulido et al. [16].
(PDF)

**S2 Table. Two-way ANOVA for the effects of temperature and $p$CO$_2$ on the number of spores released by _Sporolithon_ adults after one month in different temperature and $p$CO$_2$ treatments.** Counts of spores were taken from an n = 5–6 adults. One outlier was removed from the increased temperature treatment, which was identified using Tukey's rule and was more than 1.5 X the interquartile range.
(PDF)

## Acknowledgments

The authors thank Alexandra Ordoñez-Alvarez, Maureen Ho, Patrick Gartrell, and Ellie Bergstrom who assisted in collections and/or husbandry work. The authors would also like to thank Lizard Island Research Station for hosting us during collections.

## Author Contributions

**Conceptualization:** Tessa M. Page, Guillermo Diaz-Pulido.

**Data curation:** Tessa M. Page.

**Formal analysis:** Tessa M. Page, Guillermo Diaz-Pulido.

**Funding acquisition:** Guillermo Diaz-Pulido.

**Investigation:** Tessa M. Page.

**Methodology:** Tessa M. Page.

**Project administration:** Tessa M. Page.

**Supervision:** Guillermo Diaz-Pulido.

**Writing – original draft:** Tessa M. Page.

**Writing – review & editing:** Tessa M. Page, Guillermo Diaz-Pulido.

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
