## [Decision Letter · Decision Letter 0]

13 May 2020

PONE-D-20-07282

Plasticity of adult coralline algae to prolonged increased temperature and pCO2 exposure but reduced survival in their first generation

PLOS ONE

Dear Dr. Diaz-Pulido,

Thank you for submitting your manuscript to PLOS ONE. After careful consideration, we feel that it has merit but does not fully meet PLOS ONE’s publication criteria as it currently stands. Therefore, we invite you to submit a revised version of the manuscript that addresses the points raised during the review process.

Both reviewers identified the novelty and value in your paper. However, they both felt that some of your conclusions went beyond the inference space based on your experiments. In your revisions, please pay particular attention to moderating the speculative statements in the manuscript. Both reviewers give guidance for where you need to make changes. In addition to this, please revise the description of your experimental design - as it stands it is difficult to understand whether your replicates are independent or not. You should be careful to remove any ambiguity from the experimental design. There are some substantial changes that will need to be made to the manuscript, so I have asked for "major revisions", but I anticipate that you should be able to achieve the required changes.

We would appreciate receiving your revised manuscript by Jun 27 2020 11:59PM. To enhance the reproducibility of your results, we recommend that if applicable you deposit your laboratory protocols in protocols.io, where a protocol can be assigned its own identifier (DOI) such that it can be cited independently in the future. For instructions see: http://journals.plos.org/plosone/s/submission-guidelines#loc-laboratory-protocols

We look forward to receiving your revised manuscript.

Kind regards,

Bayden D. Russell

Academic Editor

PLOS ONE

Journal Requirements:

"This project was funded by the Australian Research Council (DP160103071) and a Griffith University’s equipment grant (GURIP)."

"This project was funded by the Australian Research Council (DP160103071) awarded

to G.D-P.

https://www.arc.gov.au/

The funders had no role in study design, data collection and analysis, decision to

publish, or preparation of the manuscript."

Reviewers' comments:

Reviewer's Responses to Questions

**Comments to the Author**

1. Is the manuscript technically sound, and do the data support the conclusions?

Reviewer #1: Partly

Reviewer #2: Partly

2. Has the statistical analysis been performed appropriately and rigorously? 

Reviewer #1: Yes

Reviewer #2: No

3. Have the authors made all data underlying the findings in their manuscript fully available?

Reviewer #1: Yes

Reviewer #2: No

4. Is the manuscript presented in an intelligible fashion and written in standard English?

Reviewer #1: Yes

Reviewer #2: Yes

5. Review Comments to the Author

Reviewer #1: Page, Tessa M., Guillermo Diaz-Pulido. Plasticity of adult coralline algae to prolonged increased temperature and pCO2 exposure but reduced survival in their first generation. Paper submitted to PLOS One

General Reviewer Comments:

An excellent experimental paper on relating elevated CO2 and temperature to growth, metabolism and survival in a coral reef coralline. It should be published. However, without additional information on the abundance and ecology of the species, and especially the comparative embryology and performance of germlings in the wild, I think generalized statements on the implications of the information should be more tempered and the potential problems inherent in the study outlined.

Several specific concerns:

1. Abstract: “Crustose coralline algae (CCA) are vital to coral reefs worldwide, providing structural integrity and inducing the settlement of important invertebrate larvae. CCA are known to be impacted by changes in their environment, both during early development and adulthood. However, long-term studies on either life history stages are lacking in the literature, therefore not allowing time to explore the acclimatory or potential adaptive responses of CCA to future global change scenarios. Here, we exposed an important, slow growing, reef building species of CCA, Sporolithon cf. durum.” In my coral reef experience, Sporolithon species are not important coral reef or algal ridge builders. Typically Sporolithon species occur in deeper water or are cave-dwelling species in reef environments. The collecting notes (“Fragments of Sporolithon (4 cm2) were collected on SCUBA using hammer and chisel at ~7 - 9 m depth. The fragments of Sporolithon collected were totally crustose and occurred in a stable, low light environment (~30 – 50 μmol photons m-2 s-1”), suggest this is so in this case. I note that the author makes no reference to the ecology of this species or of its abundance in coral reef cores. This does not greatly reduce the value of the information, but considerable caution on relating the results of this research to coral reef continued formation in future high CO2/temp seas is warranted.

2. The primary coral reef/algal ridge builders are Porolithon, Lithophyllum and Neogoniolithon, genera far removed evolutionarily from Sporolithon. I think some context as to how anatomy and cell wall structure and calcification in Sporolithon relate to the principal reef builders is desirable if the conclusions of this study are going to be related to stressors of CCA in coral reefs.

3. Intro: “CCA secrete the most soluble polymorph of CaCO3, high-Mg calcite [15] and their process of calcification is suggested to be biologically induced rather than controlled [16], and therefore are thought to be particularly sensitive to changes in seawater carbonate chemistry that occur with OA [17-21]”. Not wishing to engage the meaning of “biologically induced”, I point out that from recent published studies: CCA grow and calcify (with apparently normal wall structures) in the dark; include widely diverse parasitic genera and species lacking pigments or photosynthesis (and yet producing apparently normal calcification); develop an extraordinary range of Mg concentration (even to dolomite composition, if not crystal structure); and grow in low salinity (and in at least one species) fresh water. The wall-building and calcification processes must be metabolically driven, even if we do not know the precise chemical process. I question the general statement that “CCA are - -particularly sensitive to changes in seawater carbonate chemistry that occur with OA.”

4. “The order that Sporolithon belongs to, Sporolithales, has basal evolutionary origins having evolved and diversified during times of elevated temperature and pCO2 (relative to current levels) [35, 36], due to this, it was hypothesised that the adult Sporolithon would acclimatise more readily than their germlings to prolonged exposure to increased temperature and pCO2, resulting in low mortality and no effect on metabolic rate across all treatments.” Sporolithon is likely a relatively ancient genus, but S. durum is probably not an ancient species, as suggested. If secure paleontological information on this species and its range in geological time is available, it should be cited.

5. The experiment is well-executed and adults, spore development and release are not affected by the experimental treatments. Significant germling failure (survival, growth and metabolism) is expressed only after many weeks at high temperature and CO2. Because of this, there should have been in situ (reef) controls for the development of germlings, as well as the laboratory controls. Some information on spore settlement ecology and embryology would be helpful for the interpretation of the principal results. As earlier work on C. compactum shows, germling development in corallines is a complex process that includes break-out through the spore wall as hypothallium is developed. Is this a critical stage? As recent literature shows for some species, the basal tissue (hypothallium) has a very different wall chemistry from mature (perithallial) tissue; could this be a factor in early development of germlings. A germling starts as a single large cell, and passes through a complex embryological process to develop a range of tissues; when do the pH/T stressors apply. What stages of early development were the experimental germlings in? Also, in the natural environment, other species, especially algal competitors and grazers, can greatly influence settlement and growth, positive or negative. Substrate in the wild is likely porous carbonate. Would the underlying carbonate buffer in the wild change local sea water acidity within the germling anatomy? Perhaps more crucial, the carbonate substrate that germlings settle on is not likely a dead porous surface; it is likely quite alive at least with boring algal and blue-green filaments; even in a low light environment, it may have a multispecies algal turf. What role does this living surface play in pH control and germling development? We need to know how this surface differs from an acrylic surface relative to your treatments to feel fully secure in the primary conclusions of this study.

6. “Decreased growth rate in our Sporolithon germlings, particularly under combined high temperature and pCO2, could be linked to skeletal dissolution, as documented in other CCA species and a variety of marine calcifiers across life history stages. “ It would be helpful to have SEM based anatomical information that included comparison of calcified wall structure, since there is now considerable information in the literature on the fine structure of the calcified cell walls.

7. Finally, I think the comparisons with the performance of invertebrate species from the literature to be inappropriate for a red alga. Surely there is an alga literature (eg, Halimeda) that would be closer.

Reviewer #2: General comments

This study is interesting, mainly because it provides data on the response of germlings to climate change. Some results are interesting and useful as we need such results on numerous species to full understand changes in communities and because different species often show different responses to comparable treatments.

However, I consider thaat some flaws prevent the ready to fully understand results as well as tobe fully convinced by the interpretations. I also pointed out some speculative comments that should be removed. Moreover, despite I am not an English speaker, I feel that some parts of the text should be polished and should merit a more formal language.

I have some problems with the experimental set-up and I am not able to know if replicates are really independant, that could be a problem for the validity of statistics.

Detailed comments

L. 116: have all fragments the same shape and the same area, how did you neasure it, what about variability?

L. 126: the same, explain with more details

L. 149: Experimental setup. It is unclear. Did you use independant tanks with separate water inflows for each and separate outflows for each? It is mandatory, first to avoid contamination, second to use statistics. Maybe you should provide an diagram as supplementary material.

L. 279-280: Please explain why you performed post-hoc comparisons only when interaction was significant. It is also interesting and important to detect where are the differences among the 4 conditions even when the interaction is not significant (e.g. at weeks 7 and 8, is there only 1 value significantly different from the three other ones?

L. 293: first time you use SEM, explain. I guess it is standard error of the mean but it is also very often called SE only.

L. 344: it is a mean net O2 production, you have to precise it. The same for the Y-axis and for the caption of figure 4

L. 345: replace "between 1.74-2.30" by "between 1.74 and 2.30" and "from 4.23-3.25" by "from 3.24 to 4.23".

L. 350-351: You told the same thing with two different manners: respiration rate and the amount of O2 consumed is the same thing, would you also want to tell about O2 produced ?

L. 393-394: why a plastic or an acclimatory response ? Why not simply a sublethal stress response ?

L. 397: acclimation but L. 370: acclimatisation ?

L. 402-404: too speculative, please remove

L. 434: "Cornwall et al." instead of "Cornwall, Comeau"

L. 435-437: too spéculative, please remove

L. 447: "could have been" instead of "could' ve been", too informal

S2 Table caption: replace * by X. * corresponds to the command to multiply in numerous sofwares, including Excel, but it is not the symbol of multiplication.

6. PLOS authors have the option to publish the peer review history of their article (what does this mean?). If published, this will include your full peer review and any attached files.

Reviewer #1: Yes: Walter H. Adey

Reviewer #2: No

---

## [Author Response · Author response to Decision Letter 0]

4 Jun 2020

Dear Dr. Bayden D. Russell,

We respectfully resubmit our manuscript to PLOS ONE. We are thankful for the constructive suggestions from the reviewers, which we have incorporated into the present draft of the manuscript to the extent possible. For each reviewer’s comments/suggestions, we state each verbatim (Arial Font, highlighted in bold and italics for clarity) and then our response to each (Times Font, regular text). The authors have also added and updated references to support any changes made in response to reviewer comments. A supplemental figure, S1, has been added and Figs 3 and 4 have been updated to reflect reviewer’s comments. Additionally, we would like to add “Griffith University’s equipment grant (GURIP)” to our financial disclosure and have removed any reference to funding in the Acknowledgements section of our manuscript. 

We very much hope that you find the present version of the manuscript to be acceptable for publication.

Yours sincerely,

Guillermo Diaz-Pulido Tessa M. Page 

g.diaz-pulido@griffith.edu.au
tessa.page@griffithuni.edu.au

+ 61 7 373 53840 +61 4 247 75272 

Journal Requirements:

This has been reviewed and style requirements have been followed. 

"This project was funded by the Australian Research Council (DP160103071) and a Griffith University’s equipment grant (GURIP)."

"This project was funded by the Australian Research Council (DP160103071) awarded

to G.D-P.

https://www.arc.gov.au/

The funders had no role in study design, data collection and analysis, decision to

publish, or preparation of the manuscript."

This has been removed. 

Reviewers' comments:

5. Review Comments to the Author

Reviewer #1: Page, Tessa M., Guillermo Diaz-Pulido. Plasticity of adult coralline algae to prolonged increased temperature and pCO2 exposure but reduced survival in their first generation. Paper submitted to PLOS One

Reviewer #1: General comments

An excellent experimental paper on relating elevated CO2 and temperature to growth, metabolism and survival in a coral reef coralline. It should be published. However, without additional information on the abundance and ecology of the species, and especially the comparative embryology and performance of germlings in the wild, I think generalized statements on the implications of the information should be more tempered and the potential problems inherent in the study outlined.

The authors would like to thank Reviewer #1 for finding our manuscript to be an excellent experimental paper and for his constructive, and generally illuminating, comments and questions. We have answered each one individually below hopefully addressing them completely. We appreciate the time Reviewer #1 took to fully examine the concepts from our manuscript and provide fascinating feedback. 

Several specific concerns:

1. Abstract: “Crustose coralline algae (CCA) are vital to coral reefs worldwide, providing structural integrity and inducing the settlement of important invertebrate larvae. CCA are known to be impacted by changes in their environment, both during early development and adulthood. However, long-term studies on either life history stages are lacking in the literature, therefore not allowing time to explore the acclimatory or potential adaptive responses of CCA to future global change scenarios. Here, we exposed an important, slow growing, reef building species of CCA, Sporolithon cf. durum.” In my coral reef experience, Sporolithon species are not important coral reef or algal ridge builders. Typically Sporolithon species occur in deeper water or are cave-dwelling species in reef environments. The collecting notes (“Fragments of Sporolithon (4 cm2) were collected on SCUBA using hammer and chisel at ~7 - 9 m depth. The fragments of Sporolithon collected were totally crustose and occurred in a stable, low light environment (~30 – 50 μmol photons m-2 s-1”), suggest this is so in this case. I note that the author makes no reference to the ecology of this species or of its abundance in coral reef cores. This does not greatly reduce the value of the information, but considerable caution on relating the results of this research to coral reef continued formation in future high CO2/temp seas is warranted.

The authors acknowledge Reviewer #1’s concern with labelling Sporolithon cf. durum as an “important” reef or algal ridge builder. And, although Sporolithon can still be said to be an important structural and ecological component of tropical reefs due to its place within the CCA family, it is not a true major reef builder in comparison to other CCA species such as Porolithon or Lithophyllum. However, free-living or rhodoliths of Sporolithon durum can form extensive meadows in tropical reefs and therefore, our results have implications for reef building processes in the broader sense. To balance our statements with those from Reviewer #1, we have removed any notion of Sporolithon being a primary reef builder from our manuscript, however, we still retain their role as a reef builder. We have also mentioned its wide distribution and place within the CCA family. 

Reviewer #1 also mentioned the lack of ecological information of Sporolithon in our manuscript. Although there is some information on tropical Sporolithon in its free-living rhodolith form (e.g. Darrenougue et al. [1]; Marshall et al. [2]; Davies et al. [3]), there is very little information on this genus as a sessile, completely attached coralline alga (e.g. Verheij [4]). The authors have conducted surveys of CCA in the GBR but this information is still been processed and no quantitative trends can be drawn at this stage. However, we can say that sessile species of this genus primarily occur in low light environments (as mentioned earlier in the paper, line 140), and do not seem to be major builders of shallow-water reef frameworks, at least compared to the roles play by other CCA taxa such as Porolithon spp. or Lithophyllum spp (lines 108 - 117 & 417 – 420). We have added some additional information on the ecology of Sporolithon in the new version of the manuscript (lines 108 – 117). 

2. The primary coral reef/algal ridge builders are Porolithon, Lithophyllum and Neogoniolithon, genera far removed evolutionarily from Sporolithon. I think some context as to how anatomy and cell wall structure and calcification in Sporolithon relate to the principal reef builders is desirable if the conclusions of this study are going to be related to stressors of CCA in coral reefs.

The authors acknowledge this comment and have added statements within the manuscript clarifying that Sporolithon is not a primary reef builder, however, is still an important member of the CCA and a benthic component (as detailed before), having similar mineralisation and calcification processes as primary reef builders and it is therefore still crucial that we study this species. It is, however, worth mentioning that there are no comparative studies on the anatomy and cell wall structure and calcification of Sporolithon and major reef builders. Lines have been added in the introduction to inform the reader that although Sporolithon is not a primary reef builder, it is still important to study them – “…Sporolithon is an important reef benthic component [5] (pers. obsv) and a dominant coralline alga of rhodolith beds in the topics and subtropics, particularly in mid-to deep water environments [1-3]. Sporolithon spp. mineralise high Mg-calcite within their cell walls like the primary reef building CCA species [6]. Therefore, furthering our knowledge on the responses of Sporolithon to elevated temperature and OA [7] is not only relevant to, but vital for understanding the effects of such stressors on benthic reef communities more broadly and those species that function to cement the framework of coral reefs.” (lines 110 – 113) and in the discussion “…Although the genus Sporolithon is not a primary reef builder, they have a similar calcification process to reef building CCA species (e.g. Nash et al. [6]) and are important members of the benthic community in tropical reefs…” (lines 417 – 420).

3. Intro: “CCA secrete the most soluble polymorph of CaCO3, high-Mg calcite [15] and their process of calcification is suggested to be biologically induced rather than controlled [16], and therefore are thought to be particularly sensitive to changes in seawater carbonate chemistry that occur with OA [17-21]”. Not wishing to engage the meaning of “biologically induced”, I point out that from recent published studies: CCA grow and calcify (with apparently normal wall structures) in the dark; include widely diverse parasitic genera and species lacking pigments or photosynthesis (and yet producing apparently normal calcification); develop an extraordinary range of Mg concentration (even to dolomite composition, if not crystal structure); and grow in low salinity (and in at least one species) fresh water. The wall-building and calcification processes must be metabolically driven, even if we do not know the precise chemical process. I question the general statement that “CCA are - -particularly sensitive to changes in seawater carbonate chemistry that occur with OA.”

The authors acknowledge that this statement was quite general in regard to CCA being particularly sensitive to OA, as different taxa could, and do, respond conversely to seawater carbonate chemistry and environmental changes more broadly. A statement following this statement outlining other studies that have found species of CCA to be particularly robust and/or well adapted to extreme environments and more specifically OA, see lines 56 – 61 “In saying this, however, there are examples of CCA taxa that are well adapted to extreme environments, such as the freshwater CCA Pneophyllum cetinaensis [8], and the arctic-subarctic Clathromorphum genus [9, 10], which is able to calcify under dark conditions [11]. Moreover, some CCA taxa, particularly those from high-energy environments, have areas of the thallus which are rich in dolomite, resulting in lower dissolution rates under high-CO2 treatment [11, 12].”

4. “The order that Sporolithon belongs to, Sporolithales, has basal evolutionary origins having evolved and diversified during times of elevated temperature and pCO2 (relative to current levels) [35, 36], due to this, it was hypothesised that the adult Sporolithon would acclimatise more readily than their germlings to prolonged exposure to increased temperature and pCO2, resulting in low mortality and no effect on metabolic rate across all treatments.” Sporolithon is likely a relatively ancient genus, but S. durum is probably not an ancient species, as suggested. If secure paleontological information on this species and its range in geological time is available, it should be cited.

The authors acknowledge this comment and have included more specific citations to Peña et al. [13] and Aguirre et al. [14]. Although from the Aguirre, et al. paper [14], S. durum is suggested, from the phylogenetic tree, to having diverged around 98.5 mya from the common ancestor with the genus Heydrichia, however, as there are no other species from the Sporolithon genus represented in this tree, it is unreliable, albeit does support that the genus itself diverged around 98.5 mya, showing their basal origins. In the recently published paper by Peña, et al. (still in press) [13], the species S. durum is suggested, with notable variation, to having diverged more recently, but, the genus of Sporolithon diverged around 70 mya (although with considerable variation around the node), and the order Sporolithales around 137 mya. Both of these papers have been referenced in our manuscript. Additionally, a comment following the one under question has been added, stating that although S. cf. durum is more recently diverged, the genus still maintains its basal divergence time, lines 128 – 131. Additionally, on lines 408 & 409. 

5. The experiment is well-executed and adults, spore development and release are not affected by the experimental treatments. Significant germling failure (survival, growth and metabolism) is expressed only after many weeks at high temperature and CO2. Because of this, there should have been in situ (reef) controls for the development of germlings, as well as the laboratory controls. Some information on spore settlement ecology and embryology would be helpful for the interpretation of the principal results. As earlier work on C. compactum shows, germling development in corallines is a complex process that includes break-out through the spore wall as hypothallium is developed. Is this a critical stage? As recent literature shows for some species, the basal tissue (hypothallium) has a very different wall chemistry from mature (perithallial) tissue; could this be a factor in early development of germlings. A germling starts as a single large cell and passes through a complex embryological process to develop a range of tissues; when do the pH/T stressors apply. What stages of early development were the experimental germlings in? Also, in the natural environment, other species, especially algal competitors and grazers, can greatly influence settlement and growth, positive or negative. Substrate in the wild is likely porous carbonate. Would the underlying carbonate buffer in the wild change local sea water acidity within the germling anatomy? Perhaps more crucial, the carbonate substrate that germlings settle on is not likely a dead porous surface; it is likely quite alive at least with boring algal and blue-green filaments; even in a low light environment, it may have a multispecies algal turf. What role does this living surface play in pH control and germling development? We need to know how this surface differs from an acrylic surface relative to your treatments to feel fully secure in the primary conclusions of this study.

The authors thank Reviewer #1 for this in-depth comment and the fascinating questions asked about the spore settlement ecology of our species. Several points: 

Reviewer #1 comment on “As earlier work on C. compactum shows, germling development in corallines is a complex process that includes break-out through the spore wall as hypothallium is developed. Is this a critical stage? As recent literature shows for some species, the basal tissue (hypothallium) has a very different wall chemistry from mature (perithallial) tissue; could this be a factor in early development of germlings. A germling starts as a single large cell and passes through a complex embryological process to develop a range of tissues; when do the pH/T stressors apply. What stages of early development were the experimental germlings in?”: Our response: The germlings were in experimental seawater all the time, from their gametogenesis in the adult thalli through to spore release, settlement, germination and growth over 11 weeks (lines 240 – 244). We didn’t systematically follow the anatomical changes of spores through the process of germination and are unable to compare the cell wall anatomy of the early stages to those of the adults. We acknowledge recent research by Nash et al. [6] has found clear differences between hypothallial and perithallial cells for some species of CCA, as has another study comparing four species from the genus Phymatolithon, which found different phases of Mg-calcite in perithallial cell walls (mean 13.4 mol% MgCO3) and in the hypothallium (mean 17.1 mol% MgCO3) [15]. Nonetheless, we are unable to confirm this in our experimental Sporolithon, as this was not the focus of our study. It is however, a very interesting question and something should be investigated in future studies. The role of variable skeletal mineralogy between adults and germlings has been suggested as an explanation to differences in response between the life history stages (line 584). We have expanded on this discussion within our manuscript to incorporate the valuable comments from the Reviewer #1. See lines 588 – 598, where different composition and/or quantities of cell-wall organic constituents has been suggested as an explanation for differences in response of life history stages and text following this considering the variation of tissue organisation throughout ontogenesis. 

Reviewer #1 comment on “What role does this living surface play in pH control and germling development?” Our response: The authors have added a comment into the discussion acknowledging the potential for substrate interaction. Our previous studies have found substrate to not be critical for CCA community structure Kennedy et al. [16], however, substrate-CCA interactions and OA have not been investigated. The reviewer adds another level of complexity to the ecology of the early stages, which is the interactions with the “living substrate”, which is filled with endolithic and epilithic algae and all the associated biogeochemical process. It is well documented that the benthos exerts a strong control of the carbonate chemistry in the adjacent micro- and macro-environment, as well as on the water column (e.g. Anthony et al. [17]; McNichol et al. [18]). Therefore, conceptually, the nature of the living substrate may influence germling development and pH effects in complex ways (e.g. altering the diffusive boundary layers, photosynthesis, calcification/dissolution, grazing, etc), and this may influence spore development. This suggests that our results in a laboratory setting using an artificial substrate could potentially be different to those obtained in a natural living substrate. Unfortunately, we have not conducted a comparative study on spore development between lab and field settings and are unable to answer these questions. On the other hand, using living substrates may make the attribution of the effects of temperature/OA on germlings difficult, as there would be direct and indirect interactions with algal turfs, endolithic algae, etc, all altering the dissolution of calcium carbonate, influencing the buffering capacity of microenvironments, etc, as mentioned by Reviewer #1. As a first step in our research program, we have addressed temperature/OA on germling survival, and future studies should address complexities with living substrates. We have however acknowledged the complex interactions between living substrates and spores within the manuscript (lines 525 – 532). 

6. “Decreased growth rate in our Sporolithon germlings, particularly under combined high temperature and pCO2, could be linked to skeletal dissolution, as documented in other CCA species and a variety of marine calcifiers across life history stages. “ It would be helpful to have SEM based anatomical information that included comparison of calcified wall structure, since there is now considerable information in the literature on the fine structure of the calcified cell walls.

The authors agree that having SEM information for this study would be illuminating, and although originally intentioned, SEM images were not gathered for the germlings and therefore we cannot present this data here. 

7. Finally, I think the comparisons with the performance of invertebrate species from the literature to be inappropriate for a red alga. Surely there is an alga literature (eg, Halimeda) that would be closer.

The authors acknowledge this comment and have replaced some comparisons to other algal or seagrass species (see lines 473 – 478). However, we would like to leave a few, impactful references comparing invertebrates as PLOS ONE has a broad audience and we find the comparisons to be beneficial in broadening the reach of this manuscript and in appealing to this larger audience. 

Reviewer #2: General comments

This study is interesting, mainly because it provides data on the response of germlings to climate change. Some results are interesting and useful as we need such results on numerous species to full understand changes in communities and because different species often show different responses to comparable treatments.

The authors would like to thank Reviewer #2 for appreciating the value of our manuscript and finding aspects of it interesting. We would also like to thank Reviewer #2 for their constructive comments and hope we have addressed them fully and made appropriate edits to the manuscript. 

However, I consider thaat some flaws prevent the ready to fully understand results as well as tobe fully convinced by the interpretations. I also pointed out some speculative comments that should be removed. Moreover, despite I am not an English speaker, I feel that some parts of the text should be polished and should merit a more formal language.

We have addressed Reviewer #2’s concerns below and have further clarified the methods in the study. Additionally, speculative comments have been removed or adapted to address Reviewer #2’s concerns. Language of the manuscript has been reviewed further, polished where necessary, and introduced a more formal language where we found needed. 

I have some problems with the experimental set-up and I am not able to know if replicates are really independant, that could be a problem for the validity of statistics.

The authors have addressed this more fully in the “detailed comments” section below.

Detailed comments

L. 116: have all fragments the same shape and the same area, how did you neasure it, what about variability?

The authors thank Reviewer #2 for making this suggestion to clarify this aspect of our methods. Sporolithon fragments were collected that were around 4 cm2 in living tissue area (edited on line 138). This was visually verified when collecting, however, was not an important metric for collection. However, the surface area of each fragment that had sori was measured (which has further been explained in lines 148 – 151) and was kept as consistent as possible (~30 – 40%). We did not deem the exact size or shape of each fragment to be notable, although all fragments were crustose and maintained similar area of tissue covered in reproductive structures. 

L. 126: the same, explain with more details

The authors have addressed this with the above comment.

L. 149: Experimental setup. It is unclear. Did you use independant tanks with separate water inflows for each and separate outflows for each? It is mandatory, first to avoid contamination, second to use statistics. Maybe you should provide an diagram as supplementary material.

The authors have further clarified the experimental methods to specifically state that each of the 44 tanks were independent and water from the sumps was fed to these tanks through separate inflow tubes (lines 170, 180, 191, & 192). Additionally, a diagram of the recirculating system has been made and supplied in supplementary material (Fig S1). 

L. 279-280: Please explain why you performed post-hoc comparisons only when interaction was significant. It is also interesting and important to detect where are the differences among the 4 conditions even when the interaction is not significant (e.g. at weeks 7 and 8, is there only 1 value significantly different from the three other ones?

We thank Reviewer #2 for this comment, and this has prompted us to rethink the use of the post-hoc comparisons. We have excluded port-hoc comparisons as the ANOVAs of the maim effects of each factor are already giving us the comparisons we need for testing our hypothesis. To clarify, we conducted a two-way ANOVA, with temperature (two levels) and pCO2 (two levels), therefore no need to conduct post-hoc comparisons as there are only two levels. However, to further clarify, when the temp*CO2 interaction term was significant, we performed t-tests (lines 313 & 314) to calculate the differences between treatments (i.e. the effect of temperature within pCO2), as instructed in Prof Underwood’s, book [19].

L. 293: first time you use SEM, explain. I guess it is standard error of the mean but it is also very often called SE only.

This has been changed within the figure captions to read SE, instead of SEM. SE has also been defined in first usage. 

L. 344: it is a mean net O2 production, you have to precise it. The same for the Y-axis and for the caption of figure 4

This has been added within the manuscript on line 375 and within the axis and caption of figure 4. 

L. 345: replace "between 1.74-2.30" by "between 1.74 and 2.30" and "from 4.23-3.25" by "from 3.24 to 4.23".

This has been changed within the manuscript to read as suggested by Reviewer #2. 

L. 350-351: You told the same thing with two different manners: respiration rate and the amount of O2 consumed is the same thing, would you also want to tell about O2 produced ?

This repetition has been removed and now reads “respiration rates” only. 

L. 393-394: why a plastic or an acclimatory response ? Why not simply a sublethal stress response ?

The reviewer brings up an interesting point, however, the literature on plants indicates suppression of photosynthetic activity is an acclimatory response after exposure to elevated CO2, not a sublethal stress response. Additionally, as we did not observe any changes to adult Sporolithon health otherwise (i.e. no mortality or no changes in the pigmented/living layer), it in which sublethal stress from the literature has found, we would not consider this a sublethal stress response, unless potentially there was an energetic trade-off in response to the sublethal stress but as this was not seen in the independent elevated temperature treatment as well, it is more unlikely. We therefore use a plastic / acclamatory response for the photosynthetic response observed in adult Sporolithon. 

L. 397: acclimation but L. 370: acclimatisation ?

The authors thank Reviewer #2 for this catch, and this has been changed within the manuscript. It should read “acclimation” as this was a lab conducted study and did not occur in the field, which is when “acclimatisation or acclimatise” would be appropriate. 

L. 402-404: too speculative, please remove

The authors acknowledge this is hypothetical but believe this will provide directions for future research work and would like to keep this text. We have however reworded this comment to reflect this. Text added in line 442: “…however, this hypothesis needs experimental testing.”. 

L. 434: "Cornwall et al." instead of "Cornwall, Comeau"

This has been removed per comment below.

L. 435-437: too spéculative, please remove

The suggestion that Sporolithon germlings could exhibit similar responses to the CCA species used in Cornwall, et al., 2020 had the authors experiment been continued has been removed from the manuscript.

L. 447: "could have been" instead of "could' ve been", too informal

This has been changed within the manuscript.

S2 Table caption: replace * by X. * corresponds to the command to multiply in numerous sofwares, including Excel, but it is not the symbol of multiplication.

The authors thank Reviewer #2 for this clarification, and this has been changed within the manuscript. 

References

1. Darrenougue N, De Deckker P, Payri C, Eggins S, Fallon S. Growth and chronology of the rhodolith-forming, coralline red alga Sporolithon durum. Marine Ecology Progress Series. 2013;474:105-19.

2. Marshall J, Tsuji Y, Matsuda H, Davies P, Iryu Y, Honda N, et al. Quaternary and tertiary subtropical carbonate platform development on the continental margin of southern Queensland, Australia. Reefs and carbonate platforms in the Pacific and Indian oceans. 1998. doi: 10.1002/9781444304879.ch9.

3. Davies PJ, Braga JC, Lund M, Webster JM. Holocene deep water algal buildups on the eastern Australian shelf. Palaios. 2004;19(6):598-609.

4. Verheij E. The genus Sporolithon (Sporolithaceae fam. nov., Corallinales, Rhodophyta) from the Spermonde Archipelago, Indonesia. Phycologia. 1993;32(3):184-96. doi: 10.2216/i0031-8884-32-3-184.1.

5. Steneck RS. The ecology of coralline algal crusts convergent patterns and adaptative strategies. Annual Review of Ecology and Systematics. 1986;17:273-303.

6. Nash MC, Diaz-Pulido G, Harvey AS, Adey W. Coralline algal calcification: A morphological and process-based understanding. PLoS ONE. 2019;14(9):e0221396.

7. Cornwall CE, Comeau S, McCulloch MT. Coralline algae elevate pH at the site of calcification under ocean acidification. Global Change Biology. 2017;23(10):4245-56.

8. Žuljević A, Kaleb S, Peña V, Despalatović M, Cvitković I, De Clerck O, et al. First freshwater coralline alga and the role of local features in a major biome transition. Scientific Reports. 2016;6(1):19642.

9. Adey WH, Halfar J, Williams B. The coralline genus Clathromorphum Foslie emend. Adey: biological, physiological, and ecological factors controlling carbonate production in an arctic-subarctic climate archive. Smithsonian Contributions to the Marine Sciences. 2013;40:1-41.

10. Rahman MA, Halfar J. First evidence of chitin in calcified coralline algae: New insights into the calcification process of Clathromorphum compactum. Scientific Reports. 2014;4:6162.

11. Nash MC, Opdyke BN, Troitzsch U, Russell BD, Adey WH, Kato A, et al. Dolomite-rich coralline algae in reefs resist dissolution in acidified conditions. Nature Climate Change. 2013;3:268-72.

12. Diaz-Pulido G, Nash MC, Anthony KR, Bender D, Opdyke BN, Reyes-Nivia C, et al. Greenhouse conditions induce mineralogical changes and dolomite accumulation in coralline algae on tropical reefs. Nature Communications. 2014;5(1):1-9.

13. Peña V, Vieira C, Carlos Braga J, Aguirre J, Rösler A, Baele G, et al. Radiation of the coralline red algae (Corallinophycidae, Rhodophyta) crown group as inferred from a multilocus time-calibrated phylogeny. Mol Phylogenet Evol. 2020;150:106845.

14. Aguirre J, Perfectti F, Braga JC. Integrating phylogeny, molecular clocks, and the fossil record in the evolution of coralline algae (Corallinales and Sporolithales, Rhodophyta). Paleobiology. 2010;36(4):519-33.

15. Nash MC, Adey W. Multiple phases of mg‐calcite in crustose coralline algae suggest caution for temperature proxy and ocean acidification assessment: Lessons from the ultrastructure and biomineralization in Phymatolithon (Rhodophyta, Corallinales). J Phycol. 2017;53(5):970-84.

16. Kennedy EV, Ordoñez A, Lewis BE, Diaz-Pulido G. Comparison of recruitment tile materials for monitoring coralline algae responses to a changing climate. Marine Ecology Progress Series. 2017;569:129-44.

17. Anthony KRN, Diaz-Pulido G, Verlinden N, Tilbrook B, Andersson AJ. Benthic buffers and boosters of ocean acidification on coral reefs. Biogeosciences. 2013;10(7):4897-909.

18. McNicholl C, Koch MS, Hofmann LC. Photosynthesis and light-dependent proton pumps increase boundary layer pH in tropical macroalgae: A proposed mechanism to sustain calcification under ocean acidification. Journal of Experimental Marine Biology and Ecology. 2019;521:151208.

19. Underwood AJ. Experiments in ecology: Their logical design and interpretation using analysis of variance. Cambridge University Press1997.

---

## [Editor Report · Decision Letter 1]

10 Jun 2020

Plasticity of adult coralline algae to prolonged increased temperature and pCO2 exposure but reduced survival in their first generation

PONE-D-20-07282R1

Dear Dr. Diaz-Pulido,

We’re pleased to inform you that your manuscript has been judged scientifically suitable for publication and will be formally accepted for publication once it meets all outstanding technical requirements.

Kind regards,

Bayden D. Russell

Academic Editor

PLOS ONE

Additional Editor Comments (optional):

Thank you for doing a good job of addressing the reviewers comments and incorporating the new material that the reviewers requested.
---

## [Editor Report · Acceptance letter]

12 Jun 2020

PONE-D-20-07282R1 

Plasticity of adult coralline algae to prolonged increased temperature and *p*CO_2_ exposure but reduced survival in their first generation 

Dear Dr. Diaz-Pulido:

I'm pleased to inform you that your manuscript has been deemed suitable for publication in PLOS ONE. Congratulations! Your manuscript is now with our production department. 

Kind regards, 

on behalf of

Dr. Bayden D. Russell 

Academic Editor

PLOS ONE